# Rare but elevated incidence of hematological malignancy after clozapine use in schizophrenia: A population cohort study

Yuqi Hu[1], Le Gao[1,2,3], Lingyue Zhou[1], Wenlong Liu[1], Cuiling Wei[1], Boyan Liu[1], Qi Sun[1], Wenxin Tian[1,4], Rachel Yui Ki Chu[1,5], Song Song[1,4], Franco Wing Tak Cheng[1], Joe Kwun Nam Chan[5], Amy Pui Pui Ng[6], Heidi Ka Ying Lo[5], Krystal Chi Kei Lee[5], Wing Chung Chang[5,7], William Chi Wai Wong[6], Esther Wai Yin Chan[1,4], Ian Chi Kei Wong[1,4,8,9], Yi Chai[1,10,11‡*], Francisco Tsz Tsun Lai[1,4,6,8‡*]

1 Centre for Safe Medication Practice and Research, Department of Pharmacology and Pharmacy, Li Ka Shing Faculty of Medicine, The University of Hong Kong, Hong Kong SAR, 2 Department of Pharmacy Administration, School of Pharmacy, Xi'an Jiaotong University, Shaanxi, China, 3 Center for Drug Safety and Policy Research, Xi'an Jiaotong University, Shaanxi, China, 4 Laboratory of Data Discovery for Health (D24H), Hong Kong Science Park, Sha Tin, Hong Kong SAR, 5 Department of Psychiatry, School of Clinical Medicine, Li Ka Shing Faculty of Medicine, The University of Hong Kong, Hong Kong SAR, 6 Department of Family Medicine and Primary Care, School of Clinical Medicine, Li Ka Shing Faculty of Medicine, The University of Hong Kong, Hong Kong SAR, 7 State Key Laboratory of Brain & Cognitive Sciences, the University of Hong Kong, Hong Kong SAR, 8 Advanced Data Analytics for Medical Science (ADAMS) Limited, Hong Kong SAR, 9 Aston Pharmacy School, Aston University Birmingham, Birmingham, United Kingdom, 10 The Hong Kong Jockey Club Centre for Suicide Research and Prevention, The University of Hong Kong, Hong Kong SAR, 11 School of Public Health, Shenzhen University Medical School, Shenzhen University, Shenzhen, China

148 These authors contributed equally to this work.
‡ These authors are joint senior authors on this work.
* ychai@szu.edu.cn (YC); fttlai@hku.hk (FTTL)

**Data Availability Statement:** Data underlying the results presented in this article cannot be shared publicly because the raw data is confidential and

## Abstract

### Background

Clozapine is widely regarded as a highly efficacious psychotropic drug that is largely under-used worldwide. Recent disproportionality analyses and nationwide case-control studies suggested a potential association between clozapine use and hematological malignancy (HM). Nevertheless, the absolute rate difference is not well-established due to the absence of analytic cohort studies. The clinical significance of such a potential risk remains unclear.

### Methods and findings

We extracted data from a territory-wide public healthcare database from January 2001 to August 2022 in Hong Kong to conduct a retrospective cohort study of anonymized patients aged 18+ years with a diagnosis of schizophrenia who used clozapine or olanzapine (drug comparator with highly similar chemical structure and pharmacological mechanisms) for 90+ days, with at least 2 prior other antipsychotic use records within both groups. Weighted by inverse probability of treatment (IPTW) based on propensity scores, Poisson regression was used to estimate the incidence rate ratio (IRR) of HM between clozapine and olanzapine users. The absolute rate difference was also estimated. In total, 9,965 patients with a

not allowed for sharing in accordance with the prevailing policies of the Hospital Authority of Hong Kong. The data may be requested from Hong Kong Hospital Authority's Central Panel on Administrative Assessment of External Data Requests (https://www3.ha.org.hk/data/Provision/Submission). The R statistical programming codes are publicly available at https://doi.org/10.5281/zenodo.13871037.

**Funding:** This work is partially funded by the Laboratory of Data Discovery for Health (D24H) under AIR@InnoHK (ICKW and FTTL) administered by the Innovation and Technology Commission. The funder had no role in study design, data collection and analysis, decision to publish, or preparation of the manuscript.

**Competing interests:** ICKW received research grants from Amgen, Janssen, GSK, Novartis, Pfizer, Bayer and Bristol-Myers Squibb and Takeda, Institute for Health Research in England, European Commission, National Health and Medical Research Council in Australia, The European Union's Seventh Framework Programme for research, technological development, Research Grants Council Hong Kong and Health and Medical Research Fund Hong Kong; consulting fees from IQVIA and World Health Organization; payment for expert testimony for Appeal Court in Hong Kong; serves on advisory committees for Member of Pharmacy and Poisons Board; is a member of the Expert Committee on Clinical Events Assessment Following COVID-19 Immunization; is a member of the Advisory Panel on COVID-19 Vaccines of the Hong Kong Government; is the non-executive director of Jacobson Medical in Hong Kong; and is the founder and director of Therakind Limited (UK), Advance Data Analytics for Medical Science (ADAMS) Limited (HK), Asia Medicine Regulatory Affairs (AMERA) Services Limited and OCUS Innovation Limited (HK, Ireland and UK). EWC reported receiving grants from the National Natural Science Foundation of China during the conduct of the study; non-financial support from Wellcome Trust; grants from Research Grants Council, Research Fund Secretariat of the Health Bureau (via the Health and Medical Research Fund), National Health and Medical Research Council (Australia), Narcotics Division of the Security Bureau of the Hong Kong Special Administrative Region, Amgen, AstraZeneca, Bayer, Bristol Myers Squibb, Janssen, Pfizer, Takeda, Novartis, and RGA Reinsurance Company; and personal fees from Pfizer, AstraZeneca, Novartis, Pfizer, and the Hong Kong Special Administrative Region Hospital Authority outside the submitted work. FTTL was supported by the RGC Postdoctoral Fellowship

median follow-up period of 6.99 years (25th to 75th percentile: 4.45 to 10.32 years) were included, among which 834 were clozapine users. After IPTW, the demographic and clinical characteristics of clozapine users were comparable to those of olanzapine users. Clozapine users had a significant weighted IRR of 2.22 (95% confidence interval (CI) [1.52, 3.34]; $p <$ 0.001) for HM compared to olanzapine users. The absolute rate difference was estimated at 57.40 (95% CI [33.24, 81.55]) per 100,000 person-years. Findings were consistent across subgroups by age and sex. Sensitivity analyses all supported the robustness of the results and showed good specificity to HM but no other cancers. The main limitation of this observational study is the potential residual confounding effects that could have arisen from the lack of randomization in clozapine or olanzapine use.

## Conclusions

Absolute rate difference in HM incidence associated with clozapine is small despite a 2-fold elevated rate. Given the rarity of HM and existing blood monitoring requirements, more restrictive indication for clozapine or special warnings may not be necessary.

## Author summary

### Why was this study done?

- Preliminary epidemiologic evidence has been reported on an elevated risk of hematological malignancy (HM) associated with clozapine use.

- The clinical significance (or insignificance) of such a risk is yet to be determined.

- Analytic cohort studies are needed to estimate the rate difference, i.e., number of additional cases associated with clozapine use, in addition to rate ratios.

### What did the researchers do and find?

- Using the territory-wide electronic health records from Hong Kong public healthcare facilities, we built a retrospective cohort to compare the incidence of HM between clozapine and olanzapine users.

- Similar with previous research, we found a 2-fold elevated rate of HM in clozapine users.

- Nevertheless, the rate difference between clozapine and olanzapine users were small, i.e., approximately 1 additional case of HM in every 1,700 person-year of exposure to clozapine.

### What do these findings mean?

- The previously observed elevated risk of HM associated with clozapine is supported by territory-wide data in Hong Kong.

under the Education Bureau of the Hong Kong Special Administrative Region Government and has received research grants from the Health Bureau as well. The other authors declare no competing interest.

**Abbreviations:** CI, confidence interval; HM, hematological malignancy; IPTW, inverse probability of treatment weighting; IRR, incidence rate ratio; SMD, standardized mean difference.

- The absolute rate difference is small and, given existing risk mitigation measures for hematological abnormalities, does not necessitate additional restrictions on the indication of clozapine or special warnings.

- As the assignment of clozapine versus olanzapine use in this cohort was not randomized, there may be underlying factors that are related to both clozapine use (versus olanzapine) and HM that gave rise to the observed association but were not considered.

## Introduction

Clozapine is a second-generation antipsychotic agent widely regarded as a highly efficacious psychotropic drug, with the lowest all-cause mortality rate observed compared with other antipsychotics [1,2]. It is largely considered underused worldwide, possibly due to blood monitoring requirements [3]. Previous research has suggested a rare risk of hematological abnormalities associated with clozapine, such as agranulocytosis, eosinophilia, and thrombocytopenia [4]. In fact, recent real-world evidence on the risk of hematological malignancy (HM) associated with the use of clozapine is also accruing. A disproportionality analysis of VigiBase in 2020 provided preliminary analytical evidence on such an association [5], which necessitated further pharmacovigilance actions [6]. In 2022, a nationwide Finnish study used a population-based case-control design and suggested a 3-fold increase in the odds of HM after using clozapine compared to other antipsychotics [7]. Similarly, a recent case-control study using the United States Veteran Health Administration Database showed comparable results [8]. Due to their retrospective ascertainment of antipsychotic use, however, the comparison between specific antipsychotic agents is not without limitations. Also, although there was a cohort study design within the Finnish study, it was used to estimate the crude incidence rate of HM without multivariable analyses, i.e., 61 versus 41 per 100,000 person-year among clozapine versus other antipsychotic users. The potential rate difference is, therefore, yet to be established.

Moreover, clozapine is typically initiated at a different level of drug resistance in patients compared to most other antipsychotics [9]. In fact, it is the only drug approved by the US Food and Drug Administration for treatment-resistant schizophrenia [10]. Hence, before using clozapine, patients have likely used other antipsychotics [11], and indication bias or confounding by other drugs is highly plausible. Unfortunately, the comparison between clozapine and other drugs on a comparable timeframe was not feasible in the retrospective case-control approach to the ascertainment of drug use. Considering this potential bias, an analytic cohort study that takes into consideration previous drug use as a proxy for the level of drug resistance is needed to substantiate the findings. Of equal importance, a cohort study can provide an estimate of the potential absolute rate difference to better inform clinical decisions.

In this study, we took advantage of a territory-wide public healthcare database in Hong Kong with comprehensive linkage to various healthcare attendance records, diagnoses, and medication use among people with schizophrenia. We aimed to test for the association of clozapine use with HM in comparison with olanzapine users, adjusting for prior other antipsychotic use within both groups. Olanzapine was chosen as a comparator due to its similar chemical structure and increasing advocacy in recent years for it to serve as an alternative to clozapine for treatment-resistant schizophrenia [12,13]. We hypothesized an elevated risk of HM following the use of clozapine compared with olanzapine.

## Methods

This study is reported as per the Reporting of Studies Conducted using Observational Routinely Collected Data (RECORD) guideline (**S1 Checklist**). There was no documented protocol for the reported analysis, but it was conceptualized and planned shortly after the publication of the Finnish case-control study in 2022 [7].

### Data source

We adopted a retrospective cohort study design for this study. The Hospital Authority of Hong Kong, which is the sole provider of public inpatient services and a major provider of public outpatient services, provided strictly anonymized and de-identified electronic health records for data analysis. As all legal residents of Hong Kong are eligible for receiving services from the public sector, the database essentially covers the entire population and all territories of Hong Kong. The electronic health records were extracted from the Clinical Management System in which clinicians input routine patient attendance, diagnosis, and medication records daily, linked by a unique person identity number. Diagnoses are coded based on the International Classification of Diseases, Ninth Revision, Clinical Modifications (ICD-9-CM), while medications are coded with British National Formulary codes as well as generic names, with dosage and duration information also available. Nearly 100% of the drugs prescribed by clinicians in the public healthcare sector in Hong Kong are dispensed in the same facilities, with highly accurate records. Timestamps are available for every record entry, facilitating the delineation of the temporality of records and events. Many pharmacoepidemiologic research studies have already been published using this database [14–17], with the accuracy of diagnostic codes well established [18].

### Participants

We included anonymized patients aged 18 years or older with a diagnosis of schizophrenia (ICD-9-CM code 295) and a record of using clozapine or olanzapine for a duration of 90 days or more in January 2001 to August 2022. The first prescription of clozapine or olanzapine date was designated as the index date. Patients who (i) did not use at least 2 other antipsychotics before the index date; (ii) those who ever used both clozapine and olanzapine in their prescription history; and (iii) those who had a prior record of any cancers (including HM) before the index date were excluded. Records dating back to the year 1999 were used to execute these exclusion criteria. Patients were followed from the index date (i.e., 90 days or more clozapine or olanzapine use) until (i) the diagnosis of HM; (ii) death; (iii) 5 years after the cessation of olanzapine or clozapine use; or (iv) end of data availability (i.e., August 31, 2022), whichever came earliest.

Ethics approval for this study was obtained from the Institutional Review Board of the University of Hong Kong/Hospital Authority Hong Kong West Cluster (HKU/HA HKW IRB, reference number: UW 20–113). Informed consent was waived as a requirement for ethics approval because the data were all anonymized.

### Outcomes

Any HM diagnosis (ICD-9-CM codes 200–209, 238.4, 238.5, 238.6, 238.7, detail in **S1 Table**) as a composite measure was adopted as the outcome of this study.

### Exposure and comparator

Clozapine and olanzapine were identified by their generic names in the electronic health records. We did not consider a 7-day, or a shorter, gap between repeated prescriptions of

antipsychotics as a discontinuation given a half-life of olanzapine ranging from 21 to 54 h [19]. Clozapine was the primary exposure in this study while olanzapine was the comparator.

## Statistical analysis

To test our hypothesis, Poisson regression was used to estimate the incidence rate ratio (IRR) with 95% confidence intervals (CI) of HM between clozapine and olanzapine users, with an offset term to account for varying follow-up times for each patient. The inverse probability of treatment weighting (IPTW), using a propensity score, was applied to balance the characteristics between the clozapine and olanzapine users. The score was estimated by the logistics regression, considering covariates including age, sex, previous use of categories of antipsychotics according to their chemical structure (phenothiazines, thioxanthene derivatives, diphenylbutylpiperidine derivatives, butyrophenone derivatives, benzamides, indole derivatives, diazepines, oxazepines, thiazepines and oxepines, and other antipsychotics), prior mental disorder diagnoses (depression, bipolar disorder, and dementia), number of antipsychotic used in the past year, and immunological disease history (autoimmune disease and acquired immune deficiencies) which have been proven to associate with the HM also used as covariates [20,21]. **S1 Table** shows the ICD-9-CM codes and generic drug names used to ascertain the covariates. The standardized mean difference (SMD) was used to identify potential imbalances between clozapine and olanzapine users. Covariates with an SMD greater than 0.1 after IPTW were further adjusted in the regression model. In addition to IRRs, weighted absolute rate differences were also estimated between the 2 groups.

Subgroup analyses by age group and sex were conducted to provide specific estimates for different demographic strata. The interaction between demographic subgroups and clozapine use was tested in a combined model to examine any potential effect modification of the association across subgroups. Additionally, several sensitivity analyses were conducted to test the robustness of our results: (i) the cohort was further restricted to patients using clozapine or olanzapine for 180 days (instead of 90); (ii) patients were censored for 3 years (instead of 5) after their cessation of clozapine or olanzapine use; (iii) to enhance the comparability with other studies which may be using slightly different statistical methods than our main analysis, a multivariable Poisson regression to employed to replicate the analysis; (iv) likewise, a Cox regression with the IPTW approach was used to estimate the hazard ratio; (v) asthma (ICD-9-CM codes 493) and other cancers (ICD-9-CM codes 140–239, excluding HM) were used as negative control outcomes to check for any unmeasurable selection bias, as it shared the same potential source of bias with HM but was not causally related to antipsychotic exposure; (vi) top and lowest 1% of the weights generated with propensity scores were truncated in a repeated main analysis despite the absence of obvious outliers; (vii) we excluded one-fourth, and half of the clozapine users who had the lowest prescribed daily dose average across the duration of use in a repeated set of main analysis to observe any potential dose-response relationship; and finally, (viii) we further confined the sample to those who used 3 or more antipsychotics prior to clozapine or olanzapine use and repeated the analysis.

Similar with other observational electronic health record studies, we assumed the absence of a disease or medication if no relevant record was identified for the patients in the database. Those without basic demographic information, i.e., age and sex, would be excluded. YH and LG were independent analysts of this study to minimize errors in coding and programming. The R statistical programming environment (version 4.1.2) was used to conduct all the described analyses, with the "survminer" package used to build cumulative incidence plot. Statistical tests were two-tailed and a *P*-value of 0.05 or smaller was indicative of statistical significance. There were no deviations from or data-driven changes to our plan for the main analysis for hypothesis testing during the analysis.

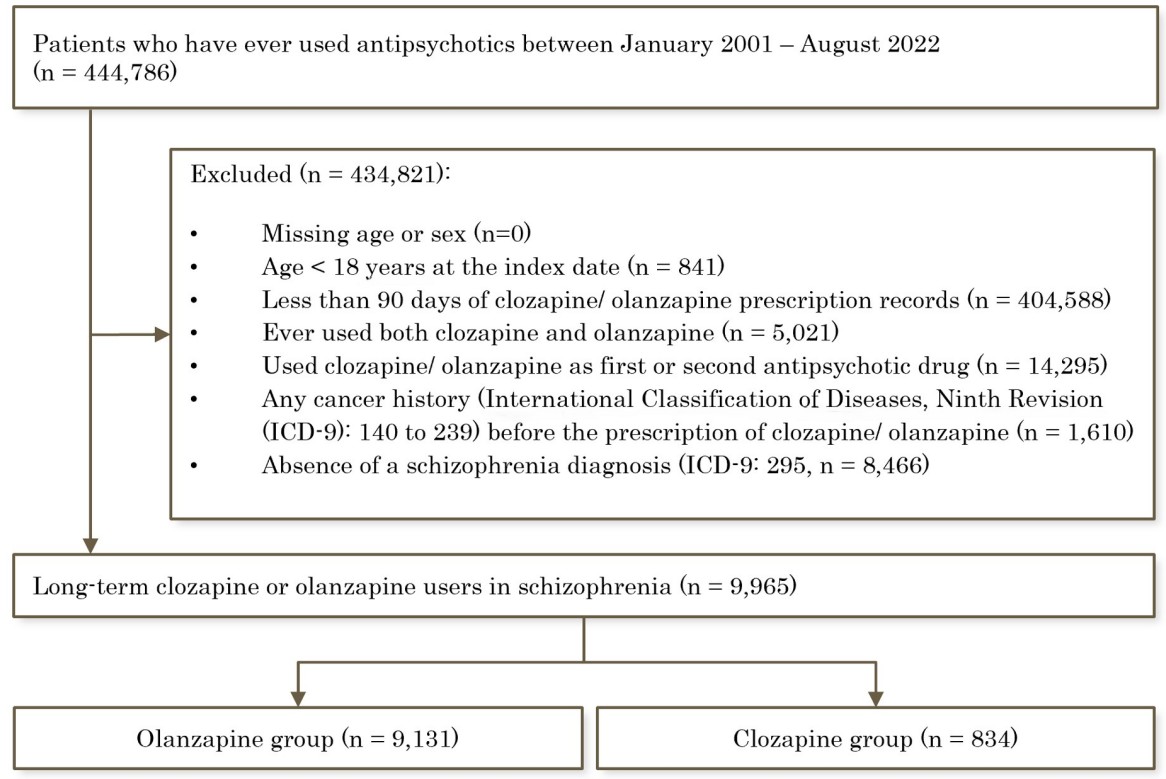

**Fig 1. Flow chart showing cohort selection processes.** ICD-9 represents International Classification of Diseases, Ninth Revision.

## Results

We identified 444,716 people from the database who had a prescription of antipsychotics between January 2001 and August 2022. After applying the exclusion criteria, the cohort size was 9,965, comprising 834 clozapine users and 9,131 olanzapine users. **Fig 1** shows the flow-chart of the cohort selection procedure.

### Cohort characteristics

**Table 1** shows the baseline characteristics of the cohort before and after applying IPTW. Before weighting, the mean age of the clozapine group (41.00 ± 15.02 years) was approximately 5 years younger than that of the olanzapine group. The olanzapine group had a higher proportion of males (54.0%) than females, while clozapine the group had a smaller proportion of males (46.6%). On average, the follow-up time of the clozapine users were 4,540.90 days compared to 2,664.43 days follow-up time in olanzapine users, and more clozapine users prescribed more than 3 kinds of other antipsychotics in the past year (59.5% versus 36.9%). Notable differences were observed in the types of previous antipsychotics use. Specifically, phenothiazines (82.6% versus 64.6%) and thioxanthene derivatives (52.8% versus 37.4%) were more often used previously among clozapine users than olanzapine users. However, more olanzapine users had used diazepines, oxazepines, thiazepines, and oxepines (39.5% versus 28.4%), and "other antipsychotics" (71.9% versus 57.3%). Additionally, a higher prevalence of psychiatric comorbidities and immunological diseases was observed among olanzapine users. After weighting, the SMD of characteristics were all less than 0.1 except prior benzamide use, which was adjusted in the subsequent weighted regression analysis.

**Table 1. Cohort characteristics.**

| | Unweighted | | | Weighted by inverse probability of treatment | | |
|---|---|---|---|---|---|---|
| | Olanzapine | Clozapine | SMD[a] | Olanzapine | Clozapine | SMD[a] |
| N | 9,131 | 834 | | 9,964.8 | 10,101.1 | |
| Mean age in years (standard deviation) | 46.48 (15.0) | 41 (11.5) | 0.404 | 46.0 (14.9) | 44.8 (12.9) | 0.084 |
| Sex (%) | | | 0.144 | | | 0.036 |
| Males | 4,932 (54) | 389 (46.6) | | 4,643 (46.6) | 4,541 (45) | |
| Females | 4,199 (46) | 445 (53.4) | | 5,321 (53.4) | 5,560 (55) | |
| Number of other antipsychotic drugs used in the past year | | | 0.515 | | | 0.024 |
| 0 | 215 (2.4) | 22 (2.6) | | 237 (2.4) | 245 (2.4) | |
| 1 | 2,203 (24.1) | 81 (9.7) | | 2,284 (22.9) | 2,229 (22.1) | |
| 2 | 3,348 (36.7) | 235 (28.2) | | 3,583 (36.0) | 3,724 (36.9) | |
| ≥3 | 3,365 (36.9) | 469 (59.5) | | 3,861 (38.7) | 3,903 (38.6) | |
| Previous other antipsychotics use (%) | | | | | | |
| Phenothiazines | 5,902 (64.6) | 689 (82.6) | 0.413 | 6,591 (66.1) | 6,493 (64.3) | 0.043 |
| Thioxanthene derivatives | 3,413 (37.4) | 440 (52.8) | 0.316 | 3,853 (38.7) | 3,638 (36) | 0.057 |
| Diphenylbutylpiperidine derivatives | 222 (2.4) | 44 (5.3) | 0.150 | 265 (2.7) | 288 (2.8) | 0.011 |
| Butyrophenone derivatives | 6,532 (71.5) | 621 (74.5) | 0.069 | 7,155 (71.8) | 7,245 (71.7) | 0.001 |
| Benzamides | 3,949 (43.2) | 364 (43.6) | 0.013 | 4,320 (43.4) | 4,903 (48.5) | 0.103 |
| Indole derivatives | 536 (5.9) | 65 (7.8) | 0.076 | 602 (6.0) | 753 (7.5) | 0.057 |
| Diazepines, oxazepines, thiazepines, and oxepines | 3,605 (39.5) | 237 (28.4) | 0.232 | 3,843 (38.6) | 4,239 (42) | 0.071 |
| Other antipsychotics | 6,563 (71.9) | 478 (57.3) | 0.312 | 7,043 (70.7) | 7,400 (73.3) | 0.059 |
| Mental illness diagnoses (%) | | | | | | |
| Depression | 1,132 (12.4) | 61 (7.3) | 0.170 | 1,194 (12.0) | 1,283 (12.7) | 0.048 |
| Bipolar disorder | 1,118 (12.2) | 73 (8.8) | 0.118 | 1,192 (12.0) | 1,362 (13.5) | 0.006 |
| Dementia | 154 (1.7) | 2 (0.2) | 0.148 | 156 (1.6) | 165 (1.6) | 0.050 |
| History of immunological diseases[b] (%) | 248 (2.7) | 18 (2.2) | 0.032 | 265 (2.7) | 191 (1.9) | 0.028 |

[a] Standardized mean difference.

[b] Immunological diseases include: (1) Autoimmune Diseases: Vitiligo, Addison disease, Alopecia areata, Autoimmune/Hashimoto's thyroiditis, Graves' disease, Morphoea, Multiple sclerosis, Myasthenia gravis, Pernicious anemia, Primary biliary cirrhosis, Takayasu arteritis, Type 1 diabetes mellitus; (2) Associated Conditions: Episcleritis/scleritis, Erythema nodosum, Haemolytic anemia, Immune thrombocytopenia purpura, Leukocytoclastic vasculitis, Myositis, Pulmonary fibrosis/interstitial lung disease, Raynaud's syndrome, Sjögren's syndrome/sicca syndrome, Thrombocytopenia purpura, Vasculitis, JIA; and (3) Acquired immune deficiencies. JIA, juvenile idiopathic arthritis; SMD, standardized mean difference.

## Main analysis

**Fig 2** shows the weighted cumulative incidence curves of HM over the follow-up period for clozapine and olanzapine users. The incidence of HM among clozapine users was consistently higher than that among olanzapine users for nearly the entire follow-up period.

Among all 9,965 individuals exposed to olanzapine or clozapine, a total of 39 individuals were diagnosed with HM. Out of the 39 cases, 9 individuals were clozapine users ($n = 834$), while the remaining 30 individuals were olanzapine users ($n = 9,131$). Six clozapine users and 23 in olanzapine users developed HM after discontinuing the medication. For the 6 clozapine users, the average number of days between their discontinuation and HM was 256.67 days. For the 23 olanzapine users, the number was 713.86 days.

Results from the Poisson regression models show that compared to olanzapine users, clozapine users had a significantly higher IRR of 2.22 (95% CI [1.52, 3.34], $p < 0.001$) for HM after applying IPTW (**Table 2**). The absolute rate difference of HM between the 2 groups was 57.40 (95% CI [33.24, 81.55]) per 100,000 person-years.

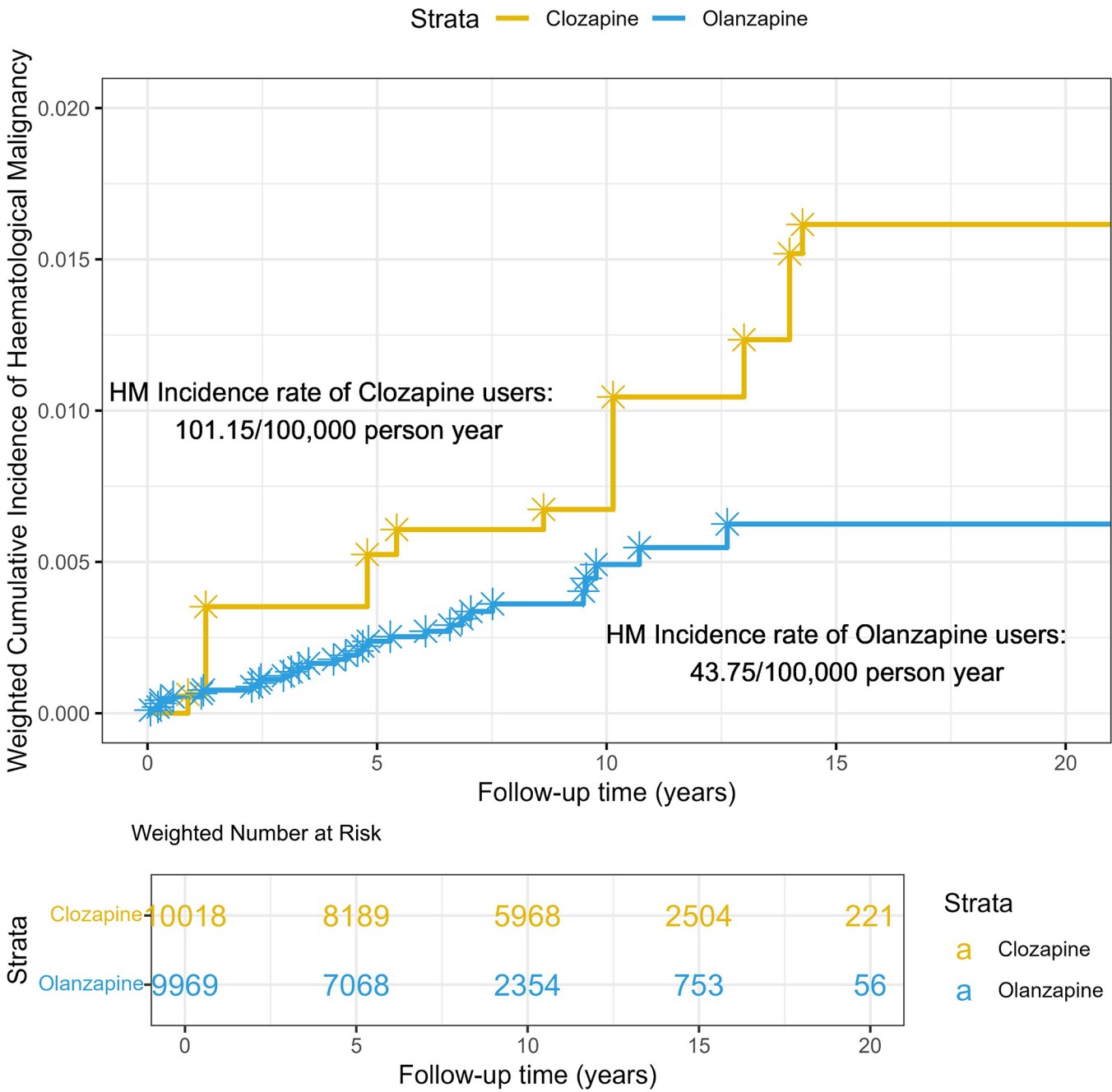

**Fig 2. Weighted cumulative incidence curves over the observation period by clozapine or olanzapine use status.** Yellow represents clozapine and blue represents olanzapine. Weighted number at risk represent the number of patients remaining in the cohort who had not experienced hematological malignancy or right-censored at different time points due to death, drug use discontinuation, or the end of data availability after inverse probability of treatment weighting.

Subgroup analyses showed a significantly elevated risk of HM among clozapine users for all sex and age groups, except for those aged 65 years or older despite a comparable weighted IRR (**Table 2**). However, we identified no significant difference in the risk increase between sexes ($p = 0.786$) or between age groups ($p = 0.743$ between those aged 45 to 64 years and those 44 years or younger; $p = 0.290$ between those aged 65 years or older and those 44 years or younger). Hence, there was no evidence of heterogeneity between sexes or across age groups.

**Table 2. IRR of HM with 95% CI.**

| | Cohort size | Average follow-up duration (days) | Number of HM cases | Crude incidence rate of HM per 100,000 person-year | Weighted incidence rate of HM per 100,000 person-year | Crude IRR [95% CI], P-value | Weighted IRR [95% CI], P-value | Weighted absolute rate difference per 100,000 person-years [95% CI] |
|---|---|---|---|---|---|---|---|---|
| Main analysis | 9,965 | 2,821.47 | 39 | 50.66 | 78.01 | | | 57.40 [33.24, 81.55] |
| Olanzapine | 9,131 | 2,664.43 | 30 | 45.04 | 43.75 | Ref. | Ref. | |
| Clozapine | 834 | 4,540.90 | 9 | 86.80 | 101.15 | 1.93 [0.86, 3.90], 0.089 | 2.22 [1.52, 3.34], <0.001 | |
| Males | 4,644 | 2,763.26 | 18 | 51.23 | 76.01 | | | 46.04 [9.57, 82.52] |
| Olanzapine | 4,199 | 2581.79 | 15 | 50.54 | 48.98 | Ref. | Ref. | |
| Clozapine | 445 | 4,475.57 | 3 | 55.02 | 95.02 | 1.09 [0.25, 3.30], 0.901 | 2.33 [1.33, 4.31], 0.004 | |
| Females | 5,321 | 2,872.28 | 21 | 50.19 | 72.26 | | | 54.27 [23.41, 85.14] |
| Olanzapine | 4,932 | 2,734.78 | 15 | 40.62 | 39.34 | Ref. | Ref. | |
| Clozapine | 389 | 4,615.64 | 6 | 122.06 | 93.61 | 3.00 [1.07, 7.39], 0.025 | 2.35 [1.38, 4.24], 0.003 | |
| Aged 44 years or younger | 4,821 | 3,196.96 | 8 | 18.96 | 27.93 | | | 23.49 [4.33, 42.64] |
| Olanzapine | 4,301 | 2,980.84 | 5 | 14.24 | 13.75 | Ref. | Ref. | |
| Clozapine | 520 | 4,984.57 | 3 | 42.27 | 37.24 | 2.97 [0.61, 12.10], 0.140 | 2.71 [1.15, 7.67], 0.043 | |
| Aged 45–64 years | 3,967 | 2,620.41 | 22 | 77.30 | 110.06 | | | 72.69 [25.06, 120.31] |
| Olanzapine | 3,674 | 2518.13 | 17 | 67.12 | 66.96 | Ref. | Ref. | |
| Clozapine | 293 | 3,902.97 | 5 | 159.7 | 139.65 | 2.38 [0.78, 6.01], 0.091 | 2.07 [1.24, 3.65], 0.008 | |
| Aged 65 years or older | 1,177 | 1,961.14 | 9 | 142.41 | 192.19 | | | 137.22 [0.00, 301.95] |
| Olanzapine | 1,156 | 1,952.16 | 8 | 129.48 | 129.34 | Ref. | Ref. | |
| Clozapine | 21 | 2,455.48 | 1 | 708.33 | 266.55 | 5.47 [0.29, 29.81], 0.110 | 1.90 [0.62, 5.78], 0.232 | |

CI, confidence interval; HM, hematological malignancy; IRR, incidence rate ratio.

## Sensitivity analysis

All sensitivity analyses except those using asthma and other cancers as negative control outcomes yielded similar results with main analysis (**S2 Table**). The weighted IRR for asthma as a negative control outcome was not significant and close to one (weighted IRR 0.90, 95% CI [0.66, 1.21]). Likewise, the weighted IRR for other cancers was estimated at 0.77 (95% CI [0.68, 0.86]). The exclusion of 25% or 50% of clozapine users who had the lowest prescribed daily dose average generated even larger weighted IRRs, which indicate a potential dose-response relationship between clozapine use and HM, i.e., 2.93 (95% CI [2.00, 4.42]) and 3.16 (95% CI [2.17, 4.78]). The median average prescribed daily dose of clozapine was 581.57 mg/day and 13.72 mg/day for olanzapine.

## Discussion

Findings from this territory-wide cohort study involving approximately 10,000 individuals with schizophrenia in Hong Kong suggested a rare but 2-fold increased rate of HM associated with clozapine compared to olanzapine. Nevertheless, the absolute rate difference is small, with less than 60 additional cases of HM per 100,000 person years of clozapine use compared with olanzapine use. Potential confounding factors, such as previous other antipsychotic use,

underlying mental illnesses, and history of immunological diseases observed at the baseline, were properly adjusted. Subgroup analyses by age and sex, as well as sensitivity analyses, showed highly robust results with a potential dose-response relationship, supporting a comparably strong and highly specific association. Our research hypothesis is therefore well supported by the data.

To the best of our knowledge, this work is the world's first analytic cohort study on this association to estimate absolute rate differences, with only 2 prior notable case-control studies conducted in Finland [7] and the United States [8]. Although the Finnish study also provided descriptive data on the incidence of HM using a cohort design, no hypothesis testing was conducted in the cohort study. Indeed, our descriptive absolute risk estimates were highly comparable with the Finnish study's descriptive cohort [7]. The rarity of HM and the relatively low prevalence of clozapine use compared to some other widely used antipsychotics have both greatly limited previous research on this association, despite compelling evidence on the association between clozapine use and a range of hematological abnormalities [22–24]. In fact, apart from this and the 2 case-control studies, existing evidence on an association with HM arose only from passive reporting systems using disproportionality analysis which may be subject to more biases than large electronic health record epidemiologic studies due to known limitations in adverse event reporting platforms and databases [6,25]. The previous case-control studies, despite the strengths of the national registry and comprehensive veteran health database, used retrospectively ascertained long-term clozapine use as the main exposure and "little to no such use" as the comparator. This approach is not without limitations, as clozapine use is most typically preceded by a range of other antipsychotic use in the individuals because it is widely regarded as the most efficacious drug available [26]. Our current study addresses this challenge by confining our cohort to those who used 2 other antipsychotics before using clozapine (or olanzapine) and employing olanzapine as an active comparator [27]. This ensures a more proper adjustment of previous antipsychotic use before clozapine or olanzapine initiation. With a cohort design, we also estimated the absolute rate difference which is highly relevant in clinical decision-making. The current study represented evidence that is one level higher up the clinical evidence pyramid compared with the existing knowledge, i.e., from case-control to cohort evidence.

The strong association observed in this and the previous 2 case-control studies [7] as well as its good coherence with previous evidence on the hematological abnormalities [28] associated with clozapine constitute a strong case that this association is likely causal. We believe the effects of blood abnormalities being induced by the prolonged use of clozapine may accumulate over the duration of use and potentially increase the risk of malignancy through a range of potential mechanisms [29]. For example, recent evidence showed that the elevated risk of agranulocytosis following clozapine use may persist well over several years, and this mechanism could potentially impact the increased risk of other blood-related disorders among clozapine users, including a higher risk of HM [30]. Although the absolute rate difference is small, a very long duration of clozapine use, which is not uncommon, may constitute a nonnegligible cumulative risk of HM. We did not have a sufficient number of cases to separately analyze the type of HM with a more significantly elevated risk; inference about the specific underlying mechanism is thus limited. Further research should investigate specific HM types. Also, the potential reasons for the weighted IRR estimated for all other cancers being significantly lower than one requires further research to investigate, which may include potential protective effect of clozapine on certain types of cancer suggested by previous cell line studies [31].

In the existing literature, there is a consensus that clozapine is highly efficacious, but largely underused worldwide [1]. Potentially life-threatening adverse effects, such as cardiovascular events, type 2 diabetes, seizure, as well as blood dyscrasias and agranulocytosis/

granulocytopenia, in addition to bothersome side effects, such as weight gain, sedation, nausea, and hypersalivation [32,33], may have contributed to the limited use of it. Nevertheless, with appropriate risk mitigation strategies in place, clozapine is widely encouraged for earlier phases of psychosis and possibly for more patients [34]. Studies have suggested its potential efficacy for patients with bipolar disorder and other mental disorders [35]. With an enhanced understanding of its pharmacological actions and safety profile in recent years, clozapine indeed has the potential to become one of the most important psychotropic drugs in the world and act as an important instrument for safeguarding population mental health in the foreseeable future. In line with this development, this study is of great importance because it determines the clinical significance of the potential risk of HM by providing reassuring data on a small, barely concerning elevated risk. Given the existing strict requirements for blood monitoring which helps early identify HM development and the rarity of HM, the risk-benefit balance for clozapine pharmacotherapies remains largely unchanged. Subject to further studies to substantiate our findings, special warnings or more restrictive indication for clozapine may not be necessary.

One of the key strengths of this study is the cohort design with a range of important covariates properly adjusted, e.g., prior antipsychotic use, providing highly useful information on the absolute rate of HM and the rate difference. Additionally, the territory-wide database with comprehensive medical records based on the same coding system and practices also confers great strengths, such as good generalizability and representativeness of the findings.

There are, nevertheless, limitations that warrant caution. First, this evidence is observational without randomization, which may entail potential selection or indication bias. Nevertheless, blood tests prior to the prescription of clozapine should have already excluded people with a higher risk of HM; such indication bias, if any, might lead to an underestimation of the observed association. Moreover, it is difficult to conduct randomized prospective studies on such a rare outcome, so future research should likely still be focused on observational studies with multiple sites or even countries. Second, although olanzapine, particularly in high doses, can be useful for some cases, it is not a suggested replacement option for clozapine in the treatment-resistance schizophrenia due to the lack of a highly comparable indication [36]. Third, there are unobserved covariates such as socioeconomic status and lifestyle factors that were not included in this study. Notably, adherence data is not available despite the high accuracy of the prescription records. Fourth, there may be a detection bias of HM for clozapine, as it is known to be associated with hematological abnormalities which are closely monitored [37]. However, since we allowed the observation to last until 5 years after discontinuation to account for symptomatic presentation, the detection bias should not be significant. Also, the association is strong and unlikely to be purely arising from detection bias, i.e., earlier detection. In fact, our findings showed that a smaller proportion of HM cases was recorded during the use of clozapine than that during the use of olanzapine. Fifth, the dosage of treatment was not standardized during the course of treatment or across individuals and is subject to the psychiatrists' clinical judgment for each individual. Sixth, the follow-up time for the olanzapine group was shorter than that of the clozapine group, but it should likely be long enough to observe and capture the study outcome, with a median follow-up time of 6.99 years (25th to 75th percentile: 4.45 to 10.32 years). Seventh, among clozapine users, many had used olanzapine before switching to clozapine in clinical practice but we excluded approximately 5,000 patients who had a history of using olanzapine either before or after starting clozapine treatment. This exclusion has limited our sample size although we arrived at a more selective sample for a fair comparison. Finally, the study population in Hong Kong is predominantly ethnic Chinese, and the generalizability of the findings to other populations needs to be tested using multinational data.

To conclude, we conducted a territory-wide retrospective cohort study in Hong Kong and identified a rare but evidently elevated risk of HM among clozapine users with schizophrenia compared with olanzapine users. With a small rate difference and existing blood monitoring requirements, special warnings or tightening of indication may not be strictly necessary or recommended. Clinicians should be aware of this consistently observed association but noting its rarity while weighing it against the benefits of clozapine pharmacotherapies.

## Supporting information

**S1 Checklist. The RECORD statement for pharmacoepidemiology (RECORD-PE) checklist of items, extended from the STROBE and RECORD statements, which should be reported in non-interventional pharmacoepidemiological studies using routinely collected health data.**
(DOCX)

**S1 Table. ICD-9-CM diagnostic codes and generic drug names used to define covariates.**
(DOCX)

**S2 Table. Sensitivity analysis: incidence rate ratios (IRR) of hematological malignancy (HM) with 95% confidence intervals (CI).**
(DOCX)

## Acknowledgments

The authors thank the Hospital Authority for the generous provision of data and gratefully acknowledge Professor Martin Roland of the University of Cambridge for his invaluable advice.

## Author Contributions

**Conceptualization:** Yuqi Hu, Francisco Tsz Tsun Lai.

**Data curation:** Yuqi Hu, Le Gao, Yi Chai.

**Formal analysis:** Yuqi Hu, Le Gao, Yi Chai, Francisco Tsz Tsun Lai.

**Funding acquisition:** Francisco Tsz Tsun Lai.

**Investigation:** Yuqi Hu, Qi Sun, Song Song.

**Methodology:** Yuqi Hu, Le Gao, Qi Sun, Yi Chai, Francisco Tsz Tsun Lai.

**Project administration:** Yuqi Hu, Francisco Tsz Tsun Lai.

**Supervision:** Francisco Tsz Tsun Lai.

**Validation:** Song Song.

**Writing – original draft:** Yuqi Hu.

**Writing – review & editing:** Lingyue Zhou, Wenlong Liu, Cuiling Wei, Boyan Liu, Qi Sun, Wenxin Tian, Rachel Yui Ki Chu, Song Song, Franco Wing Tak Cheng, Joe Kwun Nam Chan, Amy Pui Pui Ng, Heidi Ka Ying Lo, Krystal Chi Kei Lee, Wing Chung Chang, William Chi Wai Wong, Esther Wai Yin Chan, Ian Chi Kei Wong, Yi Chai, Francisco Tsz Tsun Lai.

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
