## [Editor Report · Decision Letter 0]

5 Aug 2024

Dear Dr Lai, 

Thank you for submitting your manuscript entitled "Rare but elevated incidence of hematological malignancy after clozapine use in schizophrenia: a population cohort study" for consideration by PLOS Medicine.

Your manuscript has now been evaluated by the PLOS Medicine editorial staff and I am writing to let you know that we would like to send your submission out for external peer review.

Please re-submit your manuscript within two working days, i.e. by Aug 07 2024.

Feel free to email me at atosun@plos.org or us at plosmedicine@plos.org if you have any queries relating to your submission.

Kind regards,

Alexandra Tosun, PhD

Associate Editor

PLOS Medicine

---

## [Decision Letter · Decision Letter 1]

10 Sep 2024

Dear Dr Lai,

Many thanks for submitting your manuscript "Rare but elevated incidence of hematological malignancy after clozapine use in schizophrenia: a population cohort study" (PMEDICINE-D-24-02498R1) to PLOS Medicine. The paper has been reviewed by subject experts and a statistician; their comments are included below and can also be accessed here: [LINK]

As you will see, the reviewers agree that the study is a valuable contribution, while raising some points for you to clarify and address. After discussing the paper with the editorial team and an academic editor with relevant expertise, I'm pleased to invite you to revise the paper in response to the reviewers' comments. We plan to send the revised paper to some or all of the original reviewers, and we cannot provide any guarantees at this stage regarding publication.

We ask that you submit your revision by Oct 01 2024. However, if this deadline is not feasible, please contact me by email, and we can discuss a suitable alternative.

Don't hesitate to contact me directly with any questions (atosun@plos.org). 

Best regards, 

Alexandra 

Alexandra Tosun, PhD 

Associate Editor

PLOS Medicine

atosun@plos.org

Comments from the editorial team:

The manuscript generated a lot of discussion among the editorial team. There were some concerns about the manuscript and whether it is of sufficient general interest for MED, given the rarity of the condition and treatment. We encourage you to expand on why the study will be of interest to a general health readership and have public health relevance in the introduction and discussion.

Comments from the reviewers: 

Reviewer #1: This manuscript deals with a very important clinical issue, the risk of severe adverse effect attributable to clozapine treatment. Although two previous studies have been published earlier on this issue, reporting similar results on the risk increase of hematological cancer, this study is a valuable contribution. It confirms the previous findings, but has a slightly different methodology, which made it possible to estimate the antipsychotic exposure lead prior to two treatments (clozapine vs. olanzapine) in this study. 

I have a few comments: 

1. The authors conclude in the Abstract that "Absolute rate difference in HM incidence was very small although…" Since patients are typically treated with clozapine for 10-20 years or even longer, it can be estimated that absolute rate difference of 57.4 per 100,000 persons years would be (57.4/100,000) x 15 = 0.9% during a follow-up of 15 years. This is the same cumulative incidence which has been observed for agranulocytosis. Since mortality related to hematological cancer is higher than that related to agranulocytosis (which is rather low because of early detection and growth factor treatments), it can be estimated that excess hematological cancers probably contribute more to mortality than agranulocytosis among patients treated with clozapine. So, the wording "very rare" might be reconsidered. 

2. The conclusions discuss clinical implications, stating that psychiatrists may be more inclined to try other antipsychotics before initiating clozapine. I think that the psychiatrists should do exactly the opposite: use clozapine more and earlier. The authors should acknowledge that since clozapine is associated with substantially lower risk of death (due to antisuicidal etc. effects) than other antipsychotics (i.e., it is the safest antipsychotic treatment in terms of mortality, ref. Correll et al. Mortality in people with schizophrenia: a systematic review and meta-analysis of relative risk and aggravating or attenuating factors. World Psychiatry. 2022 Jun;21(2):248-271. doi: 10.1002/wps.20994), it should be used earlier than what is currently done.

I think that early detection of the possible hematological cancers is the main implication: patients and families should be informed about warning signs, and clinicians should be vigilant (e.g., about one third of the patients developing lymphoma have anemia before diagnosis, and this could be observed in mandatory blood monitoring of patients with clozapine). 

Reviewer #2: Thanks for the opportunity to read your manuscript. My role is statistical reviewer, so I have focused on the design, data, and analysis that are presented. I have put general comments first, followed by questions relevant to a specific section of the manuscript (with a page/line reference). 

The manuscript examines if there is evidence that exposure to clozapine puts people with schizophrenia at higher risk of blood cancers. Data is the from the linked healthcare data from Hong Kong, which links diagnoses from inpatient and outpatient records, with medication dispensing records. Participants were identified from a recorded diagnosis of schizophrenia, and use of either clozapine or olanzapine for at least 90 days (first day of this period is the inception date) between 2001 and Aug 2022, with a gap of less than 7 days in coverage ignored for defining the 90-day period. Outcomes were all haematological malignancies recorded in the in and outpatient data. A lookback period of up to 1999 for previous exposure to clozapine/olanzapine, other antipsychotics, and previous HM diagnosis was used. Patients were followed until diagnosis, death, end of data, or up to 5 years from inception. T

The main analysis used inverse-probability weights, with probability derived from a logistic regression model that included age, sex, previous use of antipsychotics, mental health history, and autoimmune disease. The main analysis used Poisson regression to estimate IRR, with an offset term for differing follow-up periods. Sub-group analyses by age and sex were included, and several sensitivity analyses that considered the effect of a longer period of exposure for inclusion, shorter follow-up, a multivariable model instead of IPW, cox regression, and negative control (asthma and other cancers as outcome). The IPW, the choice of patients exposed to olanzapine as the comparator, and use of a negative control are all robust approaches for this research question. A relatively large cohort (9965 with 834 clozapine users) was selected, and good balance achieved with IP weights. There is a small but consistent difference in absolute rate of HM between olanzapine and clozapine users. The sensitivity analyses were consistent with the main results, with a small difference in absolute rate of the negative control outcomes for other cancers. 

P4, L140. Were there any extreme weights? If so, how were these dealt with?

Was there evidence of common support? This could be demonstrated with a figure showing the distribution of the propensity score by exposure group.

P4, L145. Was there any missing covariate data? If so, what approach was used?

P5, L158. What approach was used for the subgroup analyses? Stratification by subgroup variables, or an interaction term?

P5, L158. What was the rationale of the sensitivity analyses using a multivariable model, and a Cox model?

P5, L164. What software was used?

P5, L189. What approach was used to generate the IP weighted cumulative incidence curves?

P6, L206. How should the IRR of for the negative control for all cancers be interpreted? Is this likely to be due to residual confounding?

Reviewer #3: 

Reviewer Peter F.J. Schulte: 

This is an important study adding knowledge to the queston of higher risk of haemtological malignancies (HM) in clozapine users compared to other antipsychotics (in this study olanzapine). Especially the calculation of the absolute rate difference of HM is very useful for clinical decision making. The study uses a cohort design which strenghtens the findings of earlier case-control studies. I have some suggestions for additoinal analyses and some comments on the authors'conclusion. 

1)the authors state: more clozapine users were prescribed more than three kinds of other antipsychotics in the past year (59.5% versus 36.9%).

this is not astonishing since cozapine often is prescribed after numerous trials of different antipsychotics and polypharmacy. I would suggest to make an additional analysis of clozapine and olanzapine users who used three or four or five different antipsychotics prior to the use of clozapine/olanzapine. This could strengthen the point that both groups were therapy-resistant, but that the olanzapine group did not receive clozapine despite a probable indication for it. 

2) Tiihonen et al. (Lancet Psychiatry 2022) showed that clozapine use was associated with increased odds of haematological malignancies in a dose-response manner. It would strengthen the author's case of a causal relationshiop between clozpaine and HM if they could show this association too. Also, Tiihonen found that longer but not shorter use than 5 years of clozapine was associated with an increased risk. Do the authors find the same? 

I have some comments for the discussion/conclusion. The authors state: This potential slight recalibration of the risk-benefit balance may partially impact clinical decisions about long-term clozapine use. I strongly advise to put the findings in a more cautious perspective. Prescribers and patients perceive clozapine as a hazardous drug which prevents appropriate life-saving use in many patients (Verdoux et al. Prescriber and institutional barriers and facilitators of clozapine use: A systematic review, Schizophrenia Research 2018). The authors should stress that although clozapine has this higher risk of HM it is still the antipsychotic with the lowest mortality (Taipale, H., Tanskanen, A., Meht¨al¨a, J., et al., 2020. 20-year follow-up study of physical morbidity and mortality in relationship to antipsychotic treatment in a nationwide cohort of 62,250 patients with schizophrenia (FIN20). World Psychiatry 19, 61-68.) Tiihonen et al. (2022) state: 

"Although the absolute risk for haematological malignancy among patients who had used clozapine was 0·7% during the mean of 12·3 years of follow-up, the absolute risk reduction for clozapine versus other antipsychotic drugs in all-cause mortality in the same cohort was substantially greater (about 10%), in line with a previous meta-analysis." Furthermore they found an improved prognosis of HM in clozapine users (32∙9% mortality in patients with ongoing clozapine use vs 50∙7% in non-clozapine users in the investigated cohorts).

In the light of these finding the author's advices from line 261 up to 268 are not justified (first advice), certainlyl if the higher risk of individuals with significant hematological abnormalities or a family history of HM cannot be quantified. At least should the authors state that in the light of the low overall mortality of clozapine a higher risk in these groups should be weighed against the lower overall mortality and the the hgher quality adjusted life years with clozapine. This remains the realm of shared decision making. The authors state rightly that their "potential considerations need to be supported by additional evidence" adnmitting that their advice is premature. 

I suggest that the authors conclude that the absolute rate differecne vor HM between clozapine and olanzapine users is very small: (57.40 (95% CI 33.24, 81.55) per 100,000 person-years) and in the light of the very low overall mortality of clozapine in comparison to other antipsychotics is reassuring and should not lead to more restrictive indication of clozapine nor special warnings of patients of this slight additional risk. Prescribers should know this potential complication of clozapine therapy and recognize symptoms and signs. They may choose to control in addition to the granulocytes the leucocytes, differential, erythrocytes en thrombocytes once per year. 

Reviewer #4: This important study substantiates findings from earlier Finnish and US case-control studies. The study provides incidence estimates that were broadly similar to those from the Finnish cohort study (published alongside the case control study). This and the two previous case-control studies suggest a possible causal association between clozapine and haematological cancer. Although the risk of haematological cancer is rare, this association warrants further attention from clinicians and regulatory bodies. I have minor comments for the authors.

1. Abstract: I feel it is somewhat misleading to state in the background that the absolute rate difference is unclear (recommend changing to 'not well established'). Taipale et al. reported crude incidence rates of 61 cases per 100 000 person-years among clozapine users and 41 cases per 100,000 person-years among users of other antipsychotics. This is alluded to in the introduction (page 3, line 74) and discussion (page 6, line 223). I recommend that authors state these incidence rates from the Finnish study in their introduction.

2. Methods: I recommend the authors provide more data on the daily medication records extracted from the Clinical Management System (page 4, line 104). I understand these data are widely used for pharmacoepidemiological research. However, it is not clear whether the authors analysed prescribing or dispensing data. I assume that many patients were community-dwelling and prescribed clozapine in hospital outpatient clinics. How was daily medication use ascertained for these patients?

3. Methods: If I understand correctly, a patient needed to use clozapine or olanzapine for at least 90 days to be considered a user (page 4, line 120). Were patients prescribed/dispensed clozapine or olanzapine for periods less than 90 days excluded from the analyses? If so, how many patients were excluded? 

4. Results: I recommend the authors report the median clozapine and olanzapine dose over the duration of the follow-up. The Finnish study reported that the association with haematological cancer was dose-dependent (based on cumulative defined daily doses). This may be important for interpreting the association in the present study because clozapine prevalence and dose may differ between countries (e.g. see Bachmann et al. Acta Psychiatr Scand 2017). The authors acknowledge that 'dosage of treatment was not standardized during the course of treatment…' (page 8, line 293). However, it would still be interesting to know the median prescribed doses in Hong Kong.

---

* Please upload any figures associated with your paper as individual TIF or EPS files with 300dpi resolution at resubmission; please read our figure guidelines for more information on our requirements: http://journals.plos.org/plosmedicine/s/figures. While revising your submission, please upload your figure files to the PACE digital diagnostic tool, https://pacev2.apexcovantage.com/. PACE helps ensure that figures meet PLOS requirements. To use PACE, you must first register as a user. Then, login and navigate to the UPLOAD tab, where you will find detailed instructions on how to use the tool. If you encounter any issues or have any questions when using PACE, please email us at PLOSMedicine@plos.org.

* COMPETING INTEREST: All authors must declare their relevant competing interests per the PLOS policy, which can be seen here: https://journals.plos.org/plosmedicine/s/competing-interests

For authors with ties to industry, please indicate whether any of the interests has a financial stake in the results of the current study. If NO authors have competing interests, please enter: "The authors have declared that no competing interests exist."

* DATA AVAILABILITY: The Data Availability Statement (DAS) requires revision. For each data source used in your study: 

FIGURES AND TABLES

SUPPLEMENTARY MATERIAL

REFERENCES

* Where website addresses are cited, please include the complete URL and specify the date of access (e.g. [accessed: 12/06/2024]).

STUDY TYPE-SPECIFIC REQUESTS

* Abstract: Please include the study design, population and setting, number of participants, years during which the study took place (enrollment and follow up), length of follow up, and main outcome measures.

* Please ensure that the study is reported according to the RECORD guideline (available from https://www.record-statement.org) and include the completed checklist as Supporting Information. Please add the following statement, or similar, to the Methods: "This study is reported as per the Reporting of Studies Conducted using Observational Routinely-Collected Data (RECORD) guideline (S1 Checklist)." When completing the checklist, please use section and paragraph numbers, rather than page numbers.

* For all observational studies, in the manuscript text, please indicate: (1) the specific hypotheses you intended to test, (2) the analytical methods by which you planned to test them, (3) the analyses you actually performed, and (4) when reported analyses differ from those that were planned, transparent explanations for differences that affect the reliability of the study's results. If a reported analysis was performed based on an interesting but unanticipated pattern in the data, please be clear that the analysis was data driven. 

* Please state in the Methods section whether the study had a prospective protocol or analysis plan. If a prospective analysis plan (from your funding proposal, IRB or other ethics committee submission, study protocol, or other planning document written before analyzing the data) was used in designing the study, please include the relevant document(s) with your revised manuscript as a Supporting Information file to be published alongside your study and cite it in the Methods section. A legend for this file should be included at the end of your manuscript. If no such document exists, please make sure that the Methods section transparently describes when analyses were planned, and when/why any data-driven changes to analyses took place. Changes in the analysis, including those made in response to peer review comments, should be identified as such in the Methods section of the paper, with rationale.

---

## [Decision Letter · Decision Letter 2]

23 Oct 2024

Dear Dr. Lai,

Thank you very much for re-submitting your manuscript "Rare but elevated incidence of hematological malignancy after clozapine use in schizophrenia: a population cohort study" (PMEDICINE-D-24-02498R2) for review by PLOS Medicine.

Thank you for your detailed response to the editors' and reviewers' comments. I have discussed the paper with my colleagues and the academic editor, and it has also been seen again by all of the original reviewers. The changes made to the paper were mostly satisfactory to the reviewers. As such, we intend to accept the paper for publication, pending your attention to the reviewers' and editors' comments below in a further revision. When submitting your revised paper, please once again include a detailed point-by-point response to the editorial comments.

[LINK]

In revising the manuscript for further consideration here, please ensure you address the specific points made by each reviewer and the editors. In your rebuttal letter you should indicate your response to the reviewers' and editors' comments and the changes you have made in the manuscript. Please submit a clean version of the paper as the main article file. A version with changes marked must also be uploaded as a marked up manuscript file. Please also check the guidelines for revised papers at http://journals.plos.org/plosmedicine/s/revising-your-manuscript for any that apply to your paper.

We ask that you submit your revision within 1 week (Oct 30 2024). However, if this deadline is not feasible, please contact me by email, and we can discuss a suitable alternative.

Please do not hesitate to contact me directly with any questions (atosun@plos.org). If you reply directly to this message, please be sure to 'Reply All' so your message comes directly to my inbox.

We look forward to receiving the revised manuscript.   

Sincerely,

Alexandra Tosun, PhD

Associate Editor 

PLOS Medicine

plosmedicine.org

ACADEMIC EDITOR COMMENTS

I agree with the reviewers that the responses are adequate and note that the statistics reviewer has made two further helpful suggestions. The only other comment I would make is that they have overstated the safety of clozapine in their responses - this does not reflect the current view of these medications in relation to other side effects such as weight gain, metabolic syndrome, sedation, hypersalivation. These are not "unsafe" in any sense, but some may lead to increased cardiovascular risks in the longer term.

Comments from Reviewers:

Reviewer #1: The authors have addressed all issues adequately.

Reviewer #2: Thanks for the revised manuscript and responses to my original review. The revised manuscript and replies to my review have clarified most of my original review.

I would suggest for the sub-group analyses (Table 2), the most powerful way to test for heterogeneity is to use an interaction term between olanzapine/clozapine and each of the sub-group variables, rather than examine the estimated IRR per sub-group. The IRR of those aged 65 and older has a confidence interval that overlaps that of the other age-groups, which means there isn't good evidence to establish an absence of effect in this age-group compared to the others. Given the relatively small number (21) of Clozapine users in the 65 age-group this is not surprising, and it would be more robust to say that there wasn't evidence of heterogeneity according to age-group.

Reviewer #3: The authors have improved the quality of their manuscript substantially and addressed all raised points of the reviewers. This article is important for the field of psychiatry and general medicine because it investigates the risk of hematological malignancies of a certain, last resort drug, namely clozapine. They discuss the weighing of risks thoughtully.

Reviewer #4: Thank you for your comprehensive responses to the reviewers' comments. The authors have provided satisfactorily explanations to all my original comments. However, it would be useful if the authors response to comment 4.3 in the response letter was also included in the manuscript. This is because it may seem unusual to readers not familiar with the Hong Kong health system that nearly all drugs prescribed by clinicians in the public healthcare system are dispensed in public clinics and therefore recorded in clinical management system of the Health Kong Hospital Authority. If I did not know otherwise I might assume that only medications prescribed for hospital inpatients or a small sub-subset of patients who attended public-sector outpatient clinics were included in the analyses.

[LINK]

Requests from Editors:

(Please note that the comments and line numbers correspond to the line numbers in the file "ClozapineHM - revised_clean_20241001.docx" which was provided via email.)

When revising your manuscript, please keep in mind that the manuscript should be accessible to a wide audience, including readers who may not be familiar with the content of your manuscript. Also, figures and tables should be self-explanatory on a stand-alone basis, i.e. sufficient detail is required in the descriptions/legends.

DATA AVAILABILITY

Please ensure to update the data availability section in the online submission form with the details provided in lines 416-421. 

FINANCIAL DISCLOSURE 

Please ensure to update the financial disclosure section in the online submission form with the details provided in lines 374-376. 

ABSTRACT

1) l.47: When reporting age, please add a unit, such as ‘years’. Please revise throughout the Abstract and main manuscript.

2) ll.52-55: The IRR presented in the Abstract is the weighted IRR, while the participant numbers presented here are the unweighted numbers. We feel this may not be clear to readers. Please revise and clarify. A sentence such as, "After applying IPTW, the demographic and clinical characteristics of clozapine users were comparable to those of olanzapine users.”, also might be helpful.

3) l.53: “Both groups were followed up for an average of more than seven years.” – please specify by, for example, providing a median plus interquartile range.

4) l.54: Please define ‘CI’ at first use.

5) In the last sentence of the Abstract Methods and Findings section, please describe the main limitation(s) of the study's methodology.

AUTHOR SUMMARY

In the final bullet point of 'What Do These Findings Mean?', please include the main limitations of the study in non-technical language.

INTRODUCTION

l.124, please change to: “hypothesized”

METHODS AND RESULTS

1) Thank you for providing the RECORD checklist. Please replace the line numbers with paragraph numbers per section (e.g. "Methods, paragraph 1"), since the line numbers of the final published paper may be different from the line numbers in the current manuscript.

2) Figure 1: Please define ‘ICD’ in the figure description and add a unit for age.

3) Table 1: Please add a unit for age. Please define ‘immunological diseases’ below the table.

4) l.226/Table 1: Please note that the category ‘sex’ should refer to males and females.

5) Table 2: Please add a unit for age. Please note that the category ‘sex’ should refer to males and females.

6) Figure 2: We feel that it is not really clear where the numbers at risk are derived from. The numbers are not directly reflected in the top graph which might be difficult for readers to understand, and the numbers are not included in any of the tables (except for the numbers at risk at 0 years). The figure, figure description as well as the methods section provide little detail about the analysis. Please revise and also note that the title should include the word "weighted" and the type of graph. We also suggest discussing the results briefly in the results section.

7) ll.252-253: Please add a reference to the relevant table.

8) l.255, “Sensitivity analyses yielded similar results (S2 Table).” – we suggest specifying which ones.

9) ll.255-261: When discussing IRR values, please make sure that you indicate whether you are referring to the crude or weighted value since you have provided both in your analysis. Please revise throughout the Abstract and the main manuscript.

10) S2 Table: Please note that in several columns you have indicated that a p-value is included when none is provided.

DISCUSSION

1) Please remove any subheadings from the Discussion section including the ‘Conclusion’ subheading.

2) ll.369-371, please change to: “With a small rate difference and existing blood monitoring requirements, special warnings or tightening of indication may not be strictly necessary or recommended.”

REFERENCES

Please ensure that journal name abbreviations match those found in the National Center for Biotechnology Information (NCBI) databases (http://www.ncbi.nlm.nih.gov/nlmcatalog/journals), and are appropriately formatted and capitalised. For example, in reference [21], ‘The Lancet HIV’ should be ‘Lancet HIV’ or in reference [24], ‘International Journal of Psychiatry in Clinical Practice’ should be ‘Int J Psychiatry Clin Pract’.

SUPPLEMENTARY MATERIAL

In the published article, supporting information files are accessed only through a hyperlink attached to the captions. For this reason, you must list captions at the end of your manuscript file. You may include a caption within the supporting information file itself, as long as that caption is also provided in the manuscript file. Do not submit a separate caption file.

When SI files are contained with a single file:

Please label the file as ‘S1 Supporting Information’.

Please apply alphabetical labelling to each table and figure contained within the S1 file. For example, ‘Fig A’ to ‘Fig Z’ and ‘Table A’ to ‘Table Z’.

Plain text does not need to be labelled and can just be given a title as necessary. For example, ‘Statistical Analysis Plan’.

Please cite tables/figures as ‘Fig A in S1 Supporting Information’ and/or ‘Table A in S1 Supporting Information’, for example.

Please cite plain text as, ‘Statistical Analysis Plan in S1 Supporting Information’, for example.

When SI files are uploaded as separate files:

Please label tables as ‘S1 Table’ (so on) and figures as ‘S1 Fig’ (and so on).

Any additional documents (protocols/analysis plans etc.) can be labelled as ‘S1 Protocol’, for example. Please cite items as exactly as labelled.

SOCIAL MEDIA

To help us extend the reach of your research, please provide any X (formerly known as Twitter) handle(s) that would be appropriate to tag, including your own, your co-authors’, your institution, funder, or lab. Please enter in the submission form any handles you wish to be included when we post about this paper.

---

## [Editor Report · Decision Letter 3]

5 Nov 2024

Dear Dr. Lai,

Thank you very much for re-submitting your manuscript "Rare but elevated incidence of hematological malignancy after clozapine use in schizophrenia: a population cohort study" (PMEDICINE-D-24-02498R3) for review by PLOS Medicine.

We have discussed your responses with the Academic Editor and agree that their comment has not been satisfactorily addressed as you have understated the adverse effects of clozapine by describing the side effects as "tolerable". Please see the Academic Editor's comment below and ensure that you address their comment carefully and in more detail. Please revise the paper accordingly and submit the final revision within 1 week (Nov 12 2024).

Please ensure you address the specific points made by the editors. In your rebuttal letter you should indicate your response to the editors' comments and the changes you have made in the manuscript. Please submit a clean version of the paper as the main article file. A version with changes marked must also be uploaded as a marked up manuscript file. Please also check the guidelines for revised papers at http://journals.plos.org/plosmedicine/s/revising-your-manuscript for any that apply to your paper.

A reminder that when your manuscript is accepted, an uncorrected proof of your manuscript will be published online ahead of the final version, unless you've already opted out via the online submission form. If, for any reason, you do not want an earlier version of your manuscript published online or are unsure if you have already indicated as such, please let the journal staff know immediately at plosmedicine@plos.org.

If you have any questions in the meantime, please contact me directly at atosun@plos.org.

We look forward to receiving the revised manuscript.

Sincerely,

Alexandra Tosun, PhD

Associate Editor 

PLOS Medicine

plosmedicine.org

Requests from Editors:

We feel that the Academic Editor's comment has not been satisfactorily addressed as you have understated the adverse effects of clozapine by describing the side effect as "tolerable". For example, in reference [32], the authors discuss the "potentially life-threatening adverse effects which deserve special attention", which we believe is a more accurate statement and what the Academic Editor was pointing to.

Academic Editor Comments:

The response to the point about the adverse effects of clozapine is brief and rather superficial.

They now state: “In the existing literature, there is a consensus that clozapine is highly efficacious, generally safe with known tolerable metabolic and other side effects, e.g., weight gain, gastrointestinal hypomotility, etc. [32, 33], but largely underused worldwide [1], possibly due to the need for blood monitoring [3].”

(Lines 336-339, Page 9)

The authors should revise this to reflect current evidence on cardiometabolic adverse effects, which are associated with type 2 diabetes, and sedation (which is not mentioned), which is also a major concern for patients.

---

## [Editor Report · Decision Letter 4]

6 Nov 2024

Dear Dr Lai, 

On behalf of my colleagues and the Academic Editor, Seena Fazel, I am pleased to inform you that we have agreed to publish your manuscript "Rare but elevated incidence of hematological malignancy after clozapine use in schizophrenia: a population cohort study" (PMEDICINE-D-24-02498R4) in PLOS Medicine.

I appreciate your thorough responses to the reviewers' and editors' comments throughout the editorial process. We look forward to publishing your manuscript, and editorially there is only one remaining minor stylistic/presentation point that should be addressed prior to publication. We will carefully check whether the change has been made. If you have any questions or concerns regarding these final requests, please feel free to contact me at atosun@plos.org.

Please see below the minor points that we request you respond to:

1) Figure 2: Please spell out ‘IPTW’ in the figure description.

Before your manuscript can be formally accepted you will need to complete some formatting changes, which you will receive in a follow up email (including the editorial point above). Please be aware that it may take several days for you to receive this email; during this time no action is required by you. Once you have received these formatting requests, please note that your manuscript will not be scheduled for publication until you have made the required changes.

PRESS

Sincerely, 

Alexandra Tosun, PhD 

Associate Editor 

PLOS Medicine